Neutrophil-to-lymphocyte ratio: a potential supportive marker for elderly community-acquired bloodstream infections—a retrospective study

Li Peijuan 1
Zhang Jiulin 2
Chen Kun 3
Pei Qianqian 4
Ding Mengxi 5
Wang Chunmei drwangchunmei@sina.com 5
1 General Practice Ward/International Medical Center Ward, General Practice Medical Center, West China Hospital, Sichuan University , Chengdu , China
2 Department of Internal Medicine, Minzu Central Health Center , Haoba Town, Junlian County , China
3 General Department, Health Center, Jiangan Town, Jiangan County , Yibin , China
4 Department of General Practice, Aerospace Center Hospital , Beijing , China
5 Department of Intensive Care Medicine, Xuanwu Hospital Capital Medical University , Beijing , China
Farhan Faiza
Electronic publication date: 2025 Dec 8
Publication date: 2025
Volume: 13
Electronic Location ID: e20449
Received 2025 Jun 5; Accepted 2025 Oct 31
Copyright: ©2025 Li et al.
Copyright year: 2025
Copyright holder: Li et al.
License: This is an open access article distributed under the terms of the Creative Commons Attribution License, which permits unrestricted use, distribution, reproduction and adaptation in any medium and for any purpose provided that it is properly attributed. For attribution, the original author(s), title, publication source (PeerJ) and either DOI or URL of the article must be cited.
License URL: https://creativecommons.org/licenses/by/4.0/

Keywords: Neutrophil/lymphocyte ratio, Diagnosis, Bloodstream infection, Community, The elderly

Funding: The National key research and development plan 2020YFC2005403 2021 Capital General Practice Research Project 21QK08 This work was supported by the National key research and development plan (2020YFC2005403); 2021 Capital General Practice Research Project (21QK08). The funders had no role in study design, data collection and analysis, decision to publish, or preparation of the manuscript.

==============================
Background

Exploring the value of the neutrophil/lymphocyte ratio (NLR) as a potential supportive marker in community-acquired bloodstream infection in elderly patients.

Methods

By reviewing the data of inpatients in Xuanwu Hospital of Capital Medical University from January 2012 to December 2021, a total of 58 elderly patients with community-acquired bloodstream infection and 174 inpatients without infection during the same period, matched for age, sex, and concomitant diseases, were included. Clinical data were collected and analyzed statistically.

Results

There were no significant differences in age, sex, and concomitant diseases between the infected and non-infected groups. Multivariate regression analysis showed that only increased NLR among leukocyte indexes was a risk factor for bloodstream infections. The sensitivity, specificity, and area under the curve (AUC) of increased NLR to support the diagnosis of community-acquired bloodstream infections in the elderly were 0.72, 0.99, and 0.95; the optimal cut-off value is 8.0.

Conclusion

The increase in NLR was superior to the rise of traditional white blood cell and neutrophil counts. Thus, NLR can be used as an indication for the early diagnosis of community-acquired bloodstream infection in the elderly. The optimal cutoff value of NLR for supporting the diagnosis of community-acquired bloodstream infections in elderly patients was determined to be 8.0. When NLR exceeds this threshold (> 8.0), it serves as a clinical alert for community physicians, indicating a significantly elevated probability of elderly patients developing community-acquired bloodstream infections.

Introduction

Bloodstream infection (BSI) is caused by pathogenic microorganisms, including bacteria, fungi, or viruses, entering the bloodstream. This encompasses conditions such as bacteremia and sepsis. Bacteremia is diagnosed when bacteria transiently enter the bloodstream without causing typical symptoms of toxemia. Sepsis is diagnosed when pathogenic microorganisms and/or their toxins entering the blood trigger a systemic inflammatory response syndrome. BSI incidence is substantial, with community-acquired BSI (CA-BSI) representing a significant proportion of cases, particularly among the elderly. A study reported incidence rates per 1,000 hospital admissions of 4.5 for patients aged 65–74 years, 8.8 for those aged 75–84 years, and 18.1 for those aged > 85 years (Kaye et al., 2014). A meta-analysis of 72 articles estimated the case fatality rate for CA-BSI at 6.9% (Yang et al., 2010). Furthermore, Liu, CY & Tang (2018) found that CA-BSI accounted for 60.0% of BSI cases in a retrospective analysis of 735 adult patients.

However, diagnosing CA-BSI in the elderly is challenging. Symptoms are often atypical, including the absence of a corresponding fever elevation (Seigel et al., 2012). Laboratory indicators are frequently insensitive and non-specific. Consequently, early diagnosis based solely on clinical manifestations and traditional white blood cell counts is difficult (Aminzadeh & Parsa, 2011). While blood culture remains the diagnostic gold standard for BSI, its utility is limited by prolonged turnaround time and low positivity rates, hindering early diagnosis (Book et al., 2013). Molecular diagnostic techniques offer faster results and higher sensitivity but are often impractical for widespread implementation in community healthcare settings due to resource constraints (Mwaigwisya, Assiri & O’Grady, 2015).

The neutrophil-to-lymphocyte ratio (NLR) has emerged as a promising biomarker for BSI. Multiple studies confirm its diagnostic value. Research indicates that NLR levels are significantly elevated in BSI patients compared to healthy controls. An NLR threshold of 3.09 demonstrates diagnostic utility for BSI, with a sensitivity of 75.3% and specificity of 93.6% (Qu et al., 2019). NLR also shows efficacy as an early indicator of BSI in emergency department settings (sensitivity 70%, specificity 57%) (Lowsby et al., 2015). Furthermore, NLR exhibits strong predictive performance for community-acquired infections, comparable to that of positive blood culture results (de Jager et al., 2012; Cataudella et al., 2017; Manohar et al., 2018).

Given the diagnostic challenges in community settings and the limitations of current methods, identifying suitable biomarkers for early adjunctive diagnosis of CA-BSI in the elderly is a research priority. Therefore, this study aims to evaluate the potential clinical utility of NLR as a supportive diagnostic marker in elderly patients with CA-BSI.

Materials and Methods

Research design

This retrospective analysis enrolled 232 patients aged ≥ 60 years admitted to Xuanwu Hospital of Capital Medical University between January 2012 and December 2021. Participants were stratified into BSI and non-infection groups. The BSI group comprised 58 patients diagnosed with CA-BSI infection, while 174 inpatients with non-infectious diseases were matched during the same period based on age, sex, and concomitant diseases. Ethical approval was granted by the Institutional Ethics Committee of Xuanwu Hospital (Lin Yan Shen [2022] No. 131), with waived written informed consent due to the retrospective analysis of anonymized data.

Participant selection criteria

Inclusion required: (a) age ≥ 60 years and (b) confirmed CA-BSI diagnosis. Exclusion criteria encompassed: recent major surgery (within 3 months), chronic liver disease, acquired immunodeficiency syndrome, sustained glucocorticoid use (≥ 10 mg prednisone daily or equivalent for ≥ 1 month), recent radiotherapy/chemotherapy (≤ 3 months), granulocytosis, rheumatic immune diseases (e.g., systemic lupus erythematosus, rheumatoid arthritis), hematological disorders (e.g., aplastic anemia, leukemia), or hypersplenism. The diagnosis of BSI was based on the criteria established by the Ministry of Health of the People’s Republic of China, with CA-BSI defined by absence of recent healthcare facility exposure and symptom onset before or within 48 h of admission (Ministry of Health of the People’s Republic of China, 2001).

Blood culture protocol

For BSI-group patients prior to antibiotic administration, two sets of blood cultures were drawn from distinct anatomical sites. Each set contained one anaerobic and one aerobic culture bottle (BD BACTEC™ system).

Clinical data collection

General patient information included name, age, sex, and comorbid conditions (hypertension, type 2 diabetes, coronary heart disease).

For the BSI group, data encompassed:

(a) Infection manifestations: chills, fever (axillary temperature > 37 °C), sweats, fatigue, myalgia, pharyngitis, respiratory symptoms (cough, dyspnea), gastrointestinal disturbances (nausea, diarrhea), urinary complaints, neurological alterations (drowsiness, coma), rash, hepatosplenomegaly, and vital signs.

(b) Suspected infection sites.

(c) Pre-antibiotic laboratory tests: complete blood count: White blood cell (WBC), neutrophil count (NEUT), lymphocyte count (LYM), NLR; analyzed on Sysmex XS-800i with original reagents) and blood culture results (BD 9120 system).

The non-infection group was evaluated for absence of the aforementioned clinical manifestations and underwent identical blood tests. All laboratory analyses were performed by the Central Laboratory of Xuanwu Hospital, with microbial identification conducted using the VITEK 32 system (Meniere, France).

For the BSI group, blood samples were collected before the administration of any antibiotics. For the non-infection group, samples were collected upon hospital admission.

Statistical analysis

Quantitative data were summarized using means with standard deviations for normally distributed variables, or medians with interquartile ranges for non-normally distributed variables. Categorical variables were expressed as frequencies and percentages. Normality of data was assessed using the Shapiro–Wilk test.

Intergroup comparisons employed:

(a) Independent samples t-tests for normally distributed continuous data;

(b) Mann–Whitney U tests for non-normally distributed continuous data;

(c) Pearson’s χ2 tests for categorical variables.

Binary logistic regression identified factors associated with CA-BSI in elderly patients. Multivariable logistic regression models were adjusted for age, sex, and comorbidities (hypertension, type 2 diabetes, coronary heart disease).

Diagnostic performance of leukocyte indices was evaluated through receiver operating characteristic (ROC) curve analysis, determining optimal thresholds with corresponding sensitivity and specificity. Statistical significance was defined as P < 0.05 (two-tailed).

Results

Clinical and microbiological profiles

We enrolled 232 hospitalized elderly patients (≥ 60 years): 58 with CA-BSI and 174 controls admitted for non-infectious conditions. Hypertension, diabetes, and coronary heart disease prevalence did not differ significantly between groups (all P > 0.05; Table 1).

BSI-group patients exhibited diverse clinical manifestations (detailed in Tables 2–3), which were absent in control subjects. Among 58 BSI cases, monomicrobial infections were identified in 30 Gram-positive and 24 Gram-negative cases, while polymicrobial infections occurred in four cases (specifically: two with Gram-positive/fungal co-infections and two with mixed bacterial infections) (detailed in Table 4).

Microbiological analysis revealed 65 total isolates: Gram-positive pathogens accounted for 55.4% (36/65), predominantly Staphylococcus species (25 strains including eight S.hominis, five S. haemolyticus, four S. aureus, and four S. epidermidis) and Enterococcus species (eight strains comprising six E. faecium and two E. faecalis). Gram-negative isolates represented 40.0% (26/65), with major pathogens being Escherichia coli (20.0%, 13/65) and Klebsiella species (12.3%, 8/65; including five K. pneumoniae). Fungal pathogens constituted 4.6% (3/65) of isolates, identified as Candida parapsilosis, C. glabrata, and C. albicans.

Pathogen distribution analysis showed that 49 primary BSI cases yielded 55 pathogens, whereas nine secondary BSI cases originated from urinary (n = 8) or pulmonary (n = 1) sources, with full pathogen characteristics provided in Table 4.

Table 1 Baseline results of patients with bloodstream infection and non-infection.

Basic data	The bloodstream infection group	The non-infected group	p	
Number of cases (n(%))	58 (25.0%)	174 (75.0%)	–	
Age (Year, x ¯±s)	75.2 ± 8.8	74.6 ± 8.2	0.618	
Female (n(%))	28 (48.3%)	78 (44.8%)	0.648	
Concomitant disease (n(%))				
Hypertension	38 (65.5%)	109 (62.6%)	0.808	
Type 2 diabetes	33 (56.9%)	97 (55.7%)	0.980	
Coronary heart disease	21 (36.2%)	42 (24.1%)	0.087	
Notes.

Continuous variables (e.g., age) are presented as mean ± standard deviation (SD), with intergroup comparisons performed using the independent samples t-test.

Categorical variables (e.g., sex, comorbidities) are expressed as frequency and percentage (n (%)), with intergroup comparisons analyzed by χ2 test.

p-values indicate statistical significance of intergroup differences (p < 0.05 considered significant).

Table 2 Clinical manifestations of blood flow infection group.

Symptoms	Cases (composition ratio)	Symptoms	Cases (composition ratio)	
Fever (armpit temperature >37 °C)	27 (46.6%)	Lethargy	6 (10.3%)	
Cough and sputum	27 (46.6%)	Lumbago	6 (10.3%)	
Fear of cold and chills	20 (34.5%)	Celialgia	4 (6.9%)	
Nausea and vomiting	12 (20.7%)	Bladder irritation sign	4 (6.9%)	
Dyspnea	10 (17.2%)	Diarrhea	2 (3.4%)	
Altered consciousness	8 (13.8%)	Dizziness and headache	2 (3.4%)	
Chest distress	6 (10.3%)	Hematuria	1 (1.7%)	
Notes.

Categorical variables (clinical symptoms) are presented as frequency and percentage (n (%)).

This table is descriptive and does not include intergroup comparative hypothesis testing. The non-infection group had no infection-related symptoms (as per inclusion criteria); thus, intergroup comparisons were not performed.

Table 3 Clinical manifestations of primary bloodstream infection.

Primary bloodstream infection (n = 49)	Cases (composition ratio)	
Cough and sputum	25 (51.0%)	
Fever (armpit temperature > 37 °C)	24 (49.0%)	
Fear of cold and chills	19 (38.8%)	
Nausea and vomiting	11 (22.4%)	
Dyspnea	10 (20.4%)	
Altered consciousness	7 (14.3%)	
Lethargy	6 (12.2%)	
Chest distress	5 (10.2%)	
Notes.

Categorical variables (clinical symptoms) are presented as frequency and percentage (n (%)).

This table is descriptive and does not include intergroup comparative hypothesis testing.

Table 4 Distribution characteristics of pathogenic bacteria in bloodstream infection.

Pathogenic bacteria	BSI (n)	Primary BSI (n)	Secondary BSI (n)	
Gram-positive bacteria	36	36	0	
Staphylococcus hominis	8	8	0	
Enterococcus faecium	6	6	0	
Staphylococcus haemolyticus	5	5	0	
Staphylococcus aureus	4	4	0	
Staphylococcus epidermidis	4	4	0	
Staphylococcus capitis	3	3	0	
Enterococcus faecalis	2	2	0	
Staphylococcus Auris	1	1	0	
Coagulase-negative staphylococcus	1	1	0	
Cooke’s Bacteria	1	1	0	
Bacillus diphtheriformis	1	1	0	
Gram-negative bacteria	26	18	8	
E. coli	13	7	6	
Klebsiella pneumoniae	5	4	0	
Klebsiella acidogenes	2	2	0	
Pseudomonas aeruginosa	1	1	0	
Klebsiella ornithinolytica	1	1	0	
Morganella morganii	1	1	0	
Achromobacter xylosoxidans	1	0	1	
Proteus mirabilis	1	1	1	
Serratia marcescens	1	1	0	
Fungus	3	2	1	
Nearly smooth pseudohyphae	1	1	0	
Candida glabra	1	0	1	
Candida alba	1	1	0	
Notes.

Categorical variables (pathogen type) are presented as frequency and percentage (n (%)).

This table is descriptive and does not include intergroup comparative hypothesis testing.

Diagnostic utility of leukocyte indices for CA-BSI

Peripheral blood analysis revealed significantly elevated WBC counts (P < 0.01), neutrophil counts (P < 0.01), and NLR (P < 0.01), but reduced lymphocyte counts (P < 0.01) in BSI patients versus non-infected controls (Table 5).

Table 5 Comparison of leukocyte index between patients with blood stream infection and noninfection group.

Leukocyte index	Bloodstream infection group	Non-infected group	p	
WBC (×109/L) M (P25, P75)	11.2 (8.0, 15.6)	6.4 (4.9, 6.9)	<0.001	
NEUT (×109/L) M (P25, P75)	9.7 (6.2, 15.0)	4.1 (3.5, 4.8)	<0.001	
LYM (×109/L) M (P25, P75)	0.9 (0.6, 1.3)	1.6 (1.3, 1.9)	<0.001	
NLR M (P25, P75)	9.3 (5.6, 18.6)	2.7 (2.1, 3.3)	<0.001	
Notes.

WBC, White blood cell; NEUT, Neutrophil count; LYM, Lymphocyte count; NLR, Neutrophil/lymphocyte ratio.

Continuous variables (WBC, NEUT, LYM, NLR) are presented as median with interquartile range [M (P25, P75)] due to non-normal distribution. Intergroup comparisons were performed using the Mann–Whitney U test. A two-sided P-value < 0.05 was considered statistically significant.

Correlation analysis identified demographic(age), comorbidities (hypertension, type 2 diabetes and coronary heart disease), laboratory tests (WBC, NEUT, LYM, and NLR), manifestations (fever, chills and rigors and productive cough), as significant risk factors for CA-BSI (all P < 0.05). (Table 6)

Table 6 Correlation analysis of variables associated with bloodstream infection.

Variable category	Variable name	Correlation index (ρ/φ)	P-value	
Demographic	Age	0.986	<0.05	
	Sex	−0.03	0.649	
Comorbidity	Hypertension	0.377	<0.05	
	T2DM	0.693	<0.05	
	CHD	0.388	<0.05	
Laboratory	WBC	0.574	<0.05	
×109/L	NEUT	0.642	<0.05	
	LYM	−0.482	<0.05	
	NLR	0.678	<0.05	
Manifestations	Fever	0.629	<0.05	
	Chills and rigors	0.532	<0.05	
	Productive cough	0.602	<0.05	
Notes.

Abbreviations: WBC, White blood cell; NEUT, Neutrophil count; LYM, Lymphocyte count; NLR, Neutrophil/lymphocyte ratio; T2DM, type 2 diabetes; CHD, coronary heart disease; BSI.

Statistical tests: Spearman correlation coefficient (ρ) was used for continuous variables, with values ranging from −1 (strong negative correlation) to 1 (strong positive correlation). For categorical variables, phi coefficient (φ) was applied to measure association strength. Significance levels: p < 0.05 indicates statistical significance.

Univariate logistic regression identified laboratory tests (WBC, NEUT, LYM, and NLR), comorbidities (hypertension and coronary heart disease) as significant risk factors for CA-BSI (all P < 0.001). Multivariate analysis (Hosmer–Lemeshow P = 0.733) confirmed only NLR elevation independently predicted CA-BSI (OR = 3.416; 95% CI [1.861–6.268]; P < 0.001) (Tables 7–8).

Table 7 Univariate logistic regression analysis of bloodstream infection.

		B	SE	Wald	OR	95% CI	p	
Demographic	Age	0.644	83.66	0.000	1	0,3.135 ×1071	0.994	
Laboratory	WBC	0.762	0.125	36.96	2.142	1.675,2.738	<0.001	
×109/L	NEUT	1.181	0.197	35.831	3.258	2.213, 4.797	<0.001	
	LYM	−2.885	0.458	39.732	0.056	0.023, 0.137	<0.001	
	NLR	1.145	0.201	32.49	3.144	2.120, 4.661	<0.001	
Comorbidity	Hypertension	−1.787	0.328	29.65	0.167	0.088, 0.319	<0.001	
	T2DM	−23.104	7,105.181	0	0	0	0.997	
	CHD	−2.231	0.425	27.535	0.107	0.047, 0.247	<0.001	
Infection	Fever	22.928	7,735.141	0	0	0	0.998	
Manifestations	Chills and rigors	22.724	8,987.421	0	0	0	0.998	
	Productive cough	22.928	7,735.141	0	0	0	0.998	
Notes.

WBC, White blood cell; NEUT, Neutrophil count; LYM, Lymphocyte count; NLR, Neutrophil/lymphocyte ratio; T2DM, type 2 diabetes; CHD, coronary heart disease.

Univariable logistic regression was used to assess the independent association of each white blood cell parameter with bloodstream infection. Data are presented as regression coefficient (B), standard error (SE), Wald statistic, odds ratio (OR) with 95% confidence interval (CI), and P-value. Statistical significance was defined as P < 0.05.

Table 8 Multivariate logistic regression analysis of bloodstream infection.

Index	B	SE	Wald	OR	95% CI	p	
Chills and rigors	−23.623	5,233.074	0	0		0.996	
T2DM	−21.903	4,039.611	0	0		0.996	
NLR	1.228	0.31	15.727	3.416	1.861, 6.268	<0.01	
Constant	37.064	6,610.864	0	12492446376521646.000	–	0.996	
Notes.

T2DM, type 2 diabetes; NLR, neutrophils/lymphocytes ratio.

Multivariable logistic regression analysis was performed to identify independent predictors of bloodstream infection, with model fitness assessed by the Hosmer-Lemeshow test (P = 0.104); Regression coefficients (B), standard errors (SE), Wald statistics, odds ratios (OR), and 95% confidence intervals (CI) are reported; Statistical significance was defined as P < 0.05.

The model was adjusted for age, sex, and underlying comorbidities (hypertension, type 2 diabetes, coronary heart disease).

ROC curve analysis demonstrated NLR’s superior diagnostic performance for CA-BSI (AUC = 0.95, 95% CI [0.89–1.00], P < 0.05). At the optimal cutoff of 8.0, sensitivity was 0.72 and specificity 0.99, with positive/negative predictive values of 0.95 and 0.09, respectively (Youden’s index maximized; Table 9, Fig. 1).

Table 9 Neutrophil/lymphocyte ratio assists in the diagnosis of bloodstream infection.

Index	Sensibility	Specificity	Optimum cutoff value	AUC	95% CI	Positive predictive value	Negative predictive value	
WBC	0.7	0.99	9.305	0.88	0.82,0.94	0.08	0.99	
NEUT	0.8	0.96	5.87	0.93	0.88,0.98	0.03	0.99	
NLR	0.72	0.99	8	0.95	0.89,1.00	0.95	0.09	
Notes.

WBC, White blood cell; NEUT, Neutrophil count; NLR, neutrophil/lymphocyte ratio; AUC, area under the subject operating curve.

Receiver operating characteristic (ROC) curve analysis was used to evaluate the diagnostic performance of NLR for bloodstream infection. The AUC with 95% confidence interval (CI) was calculated. The optimal cut-off value was determined by maximizing the Youden index (sensitivity + specificity − 1).

Figure 1 Receiver operating characteristic curve of white blood cell parameters for early diagnosis of bloodstream infection.

Discussion

Clinical significance of early CA-BSI diagnosis in elderly

The case fatality rate of BSI in China reaches 26.8%, underscoring the critical need for early intervention (Yang et al., 2010). Current guidelines mandate intravenous antimicrobial therapy within 1 h of sepsis diagnosis (Evans et al., 2021). Although blood culture remains the diagnostic gold standard, its 15%–25% positivity rate and 24–72 h turnaround limit utility in community settings (Book et al., 2013). Molecular methods offer faster results but are cost-prohibitive for widespread community use (Mwaigwisya, Assiri & O’Grady, 2015). Thus, identifying rapid, economical diagnostic markers for elderly CA-BSI is imperative.

Atypical presentations in elderly CA-BSI

Elderly patients frequently exhibit non-classical symptoms. In this study, 53.4% lacked fever—consistent with evidence suggesting altered thermoregulation in aging. The rectal temperature cutoff for fever in elderly is >37.8 °C, yet only 8% exceed 38.5 °C (Norman, 2000). While rectal thermometry detects 86% of febrile cases versus 32% for axillary measurements (Darowski et al., 1991). However, Singler et al. (2013) noted that although rectal thermometry is more accurate, its implementation in elderly patients is often hindered by procedural difficulties and practical barriers. Zhang, W & Lu (2022) reported 135 patients with S. aureus BSI cases and found that the proportion of febrile patients was 31.1%, while the rate of cough with sputum production was even lower at 22.2%. These findings confirm that reliance on typical symptoms risks missed diagnoses.

Diagnostic utility of leukocyte indices

Limitations of WBC/NEUT

Multivariate analysis revealed no association between WBC/NEUT elevation and CA-BSI—consistent with prior studies. Cha et al. (2018) reported poor WBC diagnostic value (AUC = 0.52), influenced by non-infectious factors like malignant tumors, burns, and other non-infectious factors (Xia et al., 2019). Wang, Wang & Li (2020) investigated the utility of white blood cell indicators for predicting BSI in elderly patients with urinary tract infections. Their analysis revealed a significant difference in NEUT counts between blood culture-positive and negative groups; However, multivariate regression analysis demonstrated that elevated NEUT levels were not independently associated with BSI. Jackaman et al. (2017) observed that NEUT in elderly individuals exhibit functional impairment and diminished pathogen-killing capacity; consequently, during infections, an effective NEUT response may be compromised. Grudzinska et al. (2020) further emphasized that in severe infections such as sepsis, bone marrow release of neutrophils is suppressed, and their phagocytic activity is significantly impaired. CA-BSI in the elderly represent a severe infectious condition. Due to age-related immune decline and suppressed bone marrow function, NEUT counts may fail to rise or even decrease. Furthermore, elevated NEUT levels are influenced by diverse non-infectious factors, rendering them an unstable indicator for diagnosing CA-BSI in this population.

Superiority of NLR

In this study, multivariate analysis of BSI revealed that among white blood cell indicators, only an elevated NLR was independently associated with community-acquired BSI in the elderly. Changes in NLR reflect the dynamic equilibrium between inflammatory response and immune status. As the numerator, NEUT—innate immune cells—are mobilized from the bone marrow into peripheral blood during BSI to phagocytose pathogens (Jackaman et al., 2017). Conversely, the denominating LYM—adaptive immune cells—may undergo apoptosis or immune suppression in BSI, resulting in peripheral lymphocytopenia (Hawkins et al., 2006). Scholars have reported that during bacterial invasion, limited receptor expression on NEUT surfaces leads to the release of immature or dysfunctional neutrophils from the bone marrow, while substantial LYM migration to infection sites causes peripheral LYM depletion (Summers et al., 2010). In a retrospective analysis of CA-BSI in adults, De Jager et al. (2010) assessed the diagnostic utility of white blood cell parameters. Elevated NLR and reduced LYM were the only significant indicators; NLR demonstrated superior diagnostic sensitivity (0.77) and specificity (0.63), with an optimal cutoff of 10.

Similarly, Tan et al. (2019) retrospectively analyzed 70 adult patients with urosepsis: Neutrophils significantly increased during sepsis but showed attenuated elevation upon progression to septic shock. This suggests that while NEUT counts have diagnostic value for sepsis, their limited change in severe BSI (e.g., septic shock) reduces utility. Concurrently, rapid LYM apoptosis due to systemic immune suppression maintains NLR dynamics. Thus, elevated NLR may hold greater diagnostic significance than isolated NEUT or LYM changes in severe BSI.

Multivariate analysis confirmed NLR elevation as the sole hematologic predictor of community-acquired BSI in the elderly. Single parameters—WBC and NEUT counts—exhibit marked fluctuations influenced by non-infectious factors, while LYM primarily reflects immune status, offering incomplete diagnostic insight. NLR integrates NEUT-driven inflammation and LYM-mediated immune response, providing a stable diagnostic marker for elderly CA-BSI. Elevated NLR outperforms elevated WBC/NEUT or reduced LYM, serving as a more reliable early indicator. Setting the NLR cutoff at > 8.0 yields high sensitivity and specificity, alerting clinicians to probable CA-BSI in elderly patients.

NLR’s broader diagnostic context

Zahorec (2021) have established reference ranges for the NLR based on extensive clinical trial data: The normal threshold for NLR ranges from 1 to 2, while values for mild, moderate, and severe infections or stress responses fall within 3–7, 7–11, and 11–17, respectively. NLR demonstrates diagnostic utility with disease-specific cutoffs across various infectious conditions.

In a study by Farah et al. (2017) involving 126 patients, the mean NLR was significantly higher in hospital-acquired pneumonia (14.37) than in community-acquired pneumonia (9.39) (P < 0.01). NLR has been established as a powerful prognostic predictor in community-acquired pneumonia patients (de Jager et al., 2012), including older adults (Cataudella et al., 2017). Khanzadeh et al. (2022) analyzed 25 studies encompassing 6,410 stroke patients, revealing that post-stroke pneumonia cases exhibited substantially elevated NLR levels (mean range: 4.6–8.2) compared to non-pneumonia cases. Asik (2021) demonstrated NLR’s diagnostic value for urinary tract infections in 406 patients, identifying an optimal cutoff of 4.6. Similarly, Festa et al. (2022) reported pooled sensitivity, specificity, and AUC of 0.72, 0.74, and 0.73, respectively, for NLR in diagnosing prosthetic joint infections among 7,537 patients undergoing total hip or knee arthroplasty. Among systemic inflammatory and infectious processes, NLR, combined with CRP, served as a useful prognostic predictor in COVID-19 patients (Regolo et al., 2022). Furthermore, NLR demonstrated an inverse relationship with PaO2/FiO2, a recognized marker of respiratory failure (Regolo et al., 2023). Li et al. (2025) explored the increased diagnostic power of combining NLR with several biomarkers to diagnose BSI in children. Furthermore, NLR is an emerging marker of the relationships between the immune system and diseases (Buonacera et al., 2022).

While NLR shows partial diagnostic value for pneumonia, urinary tract infections, and postoperative joint infections, it carries limitations. Farkas (2020) caution that exogenous steroids may elevate NLR, whereas adrenal insufficiency may reduce it. Furthermore, NLR’s applicability remains unvalidated in patients with human immunodeficiency virus or active hematologic disorders (e.g., leukemia, chemotherapy). Thus, clinical interpretation of NLR for infectious diseases requires comprehensive consideration of comorbidities, medications, and auxiliary test results.

Innovations and limitations

This study establishes NLR > 8.0 as a novel, high-specificity indicator for elderly CA-BSI—surpassing traditional markers. While retrospective data inherently restrict the precision of symptom-to-sampling time assessments, our protocol adheres to best practices in early clinical diagnosis by enforcing pre-antibiotic sampling procedures. Limitations include its retrospective, single-center design and modest sample size. Future multicenter prospective studies should validate NLR’s diagnostic utility.

Conclusion

The increase in NLR was superior to the rise in traditional white blood cell and neutrophil counts. Therefore, NLR can serve as an indicator for the early diagnosis of CA-BSI in the elderly.

As a potential supportive marker for CA-BSI in elderly patients, the optimal NLR cutoff value was determined to be 8.0. An NLR exceeding this threshold (>8.0) provides a clinical alert to community physicians, signaling a significantly elevated probability of CA-BSI development in elderly patients.

Supplemental Information

Supplemental Information 1 Raw data

Supplemental Information 2 Translations for Raw Data

Supplemental Information 3 Codebook to convert raw data

Additional Information and Declarations

Competing Interests

Author Contributions

Human Ethics

Data Availability

The authors declare there are no competing interests.

Peijuan Li conceived and designed the experiments, performed the experiments, analyzed the data, prepared figures and/or tables, authored or reviewed drafts of the article, and approved the final draft.

Jiulin Zhang performed the experiments, prepared figures and/or tables, and approved the final draft.

Kun Chen performed the experiments, prepared figures and/or tables, and approved the final draft.

Qianqian Pei performed the experiments, prepared figures and/or tables, and approved the final draft.

Mengxi Ding performed the experiments, prepared figures and/or tables, and approved the final draft.

Chunmei Wang conceived and designed the experiments, prepared figures and/or tables, authored or reviewed drafts of the article, and approved the final draft.

The following information was supplied relating to ethical approvals (i.e., approving body and any reference numbers):

The research granted Xuanwu Hospital, Capital Medical University Ethical approval to carry out the study within its facilities (Ethical Application Ref: Lin Yan Shen [2022] No. 131).

The following information was supplied regarding data availability:

The raw measurements are available in the Supplementary File.

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
