# Peer review of "Neutrophil-to-lymphocyte ratio: a potential supportive marker for elderly community-acquired bloodstream infections—a retrospective study"

_PeerJ, doi:10.7717/peerj.20449_

## Round 0.1 · original submission · Major Revisions

· Academic Editor

Major Revisions

**Language Note:** The review process has identified that the English language must be improved. PeerJ can provide language editing services - please contact us at [email protected] for pricing (be sure to provide your manuscript number and title). Alternatively, you should make your own arrangements to improve the language quality and provide details in your response letter. – PeerJ Staff

Reviewer 1 ·

Basic reporting

-The manuscript must be reviewed for language by a professional service.
-I think the title must be rewritten to replace "auxiliary diagnosis" with "supportive marker/index" or anything to that effect.
-A high AUC (0.95) with modest sensitivity (0.72) is unusual.
-You cannot say "early" in a retrospective, cross-sectional study. It is unclear when the blood samples were collected.

Experimental design

-The Introduction must be made up of several paragraphs: BSI, NLR, and the aim of the study.
-The BSI diagnostic criteria were defined according to the BSI diagnostic criteria issued by the former Ministry of Health" - reference.
-Paragraphs in scientific papers are not written in numbered bullet point (e.g., methods, conclusion... etc.).

Validity of the findings

-What normality test was used?
-If the authors say NLR is better than WBC count then Figure 1 should show WBC curve as well. The Figure must also show AUC, P value, sensitivity, specificity, PPV, and NPV. Basically the same data in Table 8 and more.
-Every table must have a footnote explaining abbreviations and statistical tests used.
-Why was data in Table 2 not presented as a comparison between the two groups?
-Table 4 should be presented earlier.
-Table 7 must show this model was adjusted for what variables.
-The authors must perform correlation analysis of BSI with demographic, lab, and clinical variables and the findings must be presented in a Table before the regression data.

Reviewer 2 ·

Basic reporting

Li et al. in this paper titled “The value of neutrophil-to-lymphocyte ratio in the auxiliary diagnosis of elderly community-acquired bloodstream infections a retrospective study” tested the diagnostic power fo NLR in community acquired systemic infections. Besides NLR, Authors showed data on Leukocyte index, but a comparison with other indexes (PLR, MLR, SII and so on) would provide further insights to strengthen the individual diagnostic power of emerging tools in this field. Overall, NLR should probably be considered the most popular and widely tested tool in several diseases somehow linked to an inflammatory process (see for reference Buonacera A. et al., Int J Mol Sci 2022).

Experimental design

Authors tested the diagnostic power of NLR in community acquired systemic infections.

Validity of the findings

Authors should evaluate their findings considering the following items 1 and 2, while items 3 and 4 are suggestions for assessing further results from their data.

1. NLR was found to be a poweful prognostic predictor in patients with community acquired pneumonia (CAP) (De Jager C.P. et al. Plos One 2012, even in older patients (Cataudella E. et al., J Am Geriatr Soc 2017). These 2 references should be commented and acknowledged.
2. Among inflammatory and infectious systemic processes, NLR was helpful together with CRP as prognostic predictor in Covid-19 patients (Regolo M. et al., J Clin Med 2022), with an inverse relationship with PaO2/FiO2, a recognized marker of respiratory failure (Regolo M. et al., J Clin Med 2023). In these patients, PaO2/FiO2 behaviour is under control of the interplay between age, neutrophils, lymhocytes and CRP (Regolo M. et al., J Clin Med 2023). Please comment and acknowledge these 2 references.
3. Can Authors provide ROC curve for Leukocyte index?
4. Can Authors provide any comparisons of NLR with other well recognized inflammatory biomarkers?

Additional comments

·

Basic reporting

This research report d'elsa with a sample of >60 yrs. patients with blood infections, but the data on 58 cases are not split further in other age groups. Different pathogenic bacteria were responsible, and we do not know which treatment the patients received and what was the outcome. Sex and other variables are not specified, since responses to sepsis may differ according to sex. The bottom line is that neutrophils increase, some cases show lymphocyte decreased but NLR fared much better to classify cases with bloodstream bacterial infections, and a cut-off of 8 was proposed, despite a less pronounced inflammatory neutrophil response in agenzia subjects

Experimental design

Small sample from a single center may be based, it should be acknowledged

Validity of the findings

Not very informative due to small sample side and lack of comparison with other indexes.

Additional comments

Not very new, it confirm previous findings.

---

## Round 0.2 · Minor Revisions

· Academic Editor

Minor Revisions

Reviewer 1 ·

Basic reporting

The authors have adequately addressed my comments. However, correlation analysis in Table 6 was done using Pearson correlation despite the authors stating the data were skewed as revealed by the Shapiro-Wilk test. Therefore, this must be repeated using Spearman's test.

I would highly encourage the authors to improve their knowledge of statistics for their future studies.

Experimental design

-

Validity of the findings

-

Reviewer 2 ·

Basic reporting

Literature references provided sufficient field background/context.

Professional article structure, figures, and tables. Raw data shared.

Self-contained with relevant results to hypotheses.

Experimental design

Original primary research within the Aims and Scope of the journal.

Research question well defined, relevant & meaningful. It is stated how research fills an identified knowledge gap.

Rigorous investigation performed to a high technical & ethical standard.

Methods described with sufficient detail & information to replicate.

Validity of the findings

Impact and novelty assessed.

All underlying data have been provided; they are robust, statistically sound, & controlled.

Conclusions are well stated, linked to the original research question & limited to supporting results.

Additional comments

Lines 94-96 (track changes version) should incorporate references by de Jager 2012 and Cataudella 2017. This is because these references were published earlier than the one cited (Manohar et al., 2018).

Reference by Li et al. 2025 (see my previous report) is missing in the reference list and must be both acknowledged and commented.

---

## Round 0.3 · Minor Revisions

· Academic Editor

Minor Revisions

**Language Note:** When preparing your next revision, please ensure that your manuscript is reviewed either by a colleague who is proficient in English and familiar with the subject matter, or by a professional editing service. PeerJ offers language editing services; if you are interested, you may contact us at [email protected] for pricing details. Kindly include your manuscript number and title in your inquiry. – PeerJ Staff

Reviewer 2 ·

Basic reporting

Clear, unambiguous, providing background context.

Experimental design

Original research, rigorously carried out. Methods are clearly described.

Validity of the findings

Meaningful conclusions.

Additional comments

As regards the missing references, contributing to focus on the paper's subject:

1) Buonacera A. et al. (Int J Mol Sci 2022) should be acknowledged, as requested in one of my previous reports, to emphasize the multitasking diagnostic potential of NLR.

2) Li Y. et al. BMC Pediatrics 2025 should be acknowledged because it explored in children the increased diagnostic power of combining several biomarkers to diagnose bloodstream infections. This Reviewer understands that retrospective expansion of biomarker datasets is methodologically unfeasible in the present study, but this is not in contrast with discussing and acknowledging in the Discussion data from other Authors, fitting with the purpose, even to correctly drive future research.

---

## Round 0.4 · accepted · Accept

· Academic Editor

Accept

Thank you for carefully addressing all the raised points in your revision. I am pleased to confirm that your article is now accepted for publication. Congratulations on your excellent work!

Reviewer 2 ·

Basic reporting

Clear and unambiguous.

Experimental design

Original primary research.

Validity of the findings

All underlying data have been provided; Conclusions are well stated.

Additional comments

No further concern.